# Diabatic heating governs the seasonality of the Atlantic Niño

Hyacinth C. Nnamchi[1,2 ✉], Mojib Latif[1], Noel S. Keenlyside [3,4], Joakim Kjellsson [1] & Ingo Richter[5]

The Atlantic Niño is the leading mode of interannual sea-surface temperature (SST) variability in the equatorial Atlantic and assumed to be largely governed by coupled ocean-atmosphere dynamics described by the Bjerknes-feedback loop. However, the role of the atmospheric diabatic heating, which can be either an indicator of the atmosphere's response to, or its influence on the SST, is poorly understood. Here, using satellite-era observations from 1982–2015, we show that diabatic heating variability associated with the seasonal migration of the Inter-Tropical Convergence Zone controls the seasonality of the Atlantic Niño. The variability in precipitation, a measure of vertically integrated diabatic heating, leads that in SST, whereas the atmospheric response to SST variability is relatively weak. Our findings imply that the oceanic impact on the atmosphere is smaller than previously thought, questioning the relevance of the classical Bjerknes-feedback loop for the Atlantic Niño and limiting climate predictability over the equatorial Atlantic sector.

[1] GEOMAR Helmholtz Centre for Ocean Research Kiel, Kiel, Germany. [2] Department of Geography, University of Nigeria, Nsukka, Nigeria. [3] Geophysical Institute, University of Bergen, and Bjerknes Centre for Climate Research, Bergen, Norway. [4] Nansen Environmental and Remote Sensing Center, Bergen, Norway. [5] Application Laboratory, Research Institute for Value-Added-Information Generation, Japan Agency for Marine-Earth Science and Technology, Yokohama, Japan. ✉email: hnnamchi@geomar.de

The interannual climate variability in the equatorial Atlantic region is dominated by the Atlantic Niño, also referred to as the zonal mode. It exhibits a characteristic zonally asymmetric structure in sea surface temperature (SST) and wind stress fluctuations[1–3]. Previous work suggested that SST-anomaly growth during the Atlantic Niño is largely governed by the Bjerknes-feedback loop, a positive feedback between adjustments in oceanic and atmospheric circulations[1–11], which is the dominant growth mechanism during the warm and cold phases of the El Niño/Southern Oscillation (ENSO) in the equatorial Pacific[1,2,5].

According to the Bjerknes-feedback loop[12–14], an initial SST anomaly in the eastern equatorial Atlantic alters the zonal SST gradient ($dSST/dx$), which in turn modifies the vertical profile of the atmospheric diabatic heating through changes in convection, water vapor, cloud cover, and precipitation[15,16] across the equatorial Atlantic. The net effect is an increase in the vertical gradient of the diabatic heating ($\Delta Q$), defined here as the difference between the mid- and lower-troposphere ($dSST/dx \rightarrow \Delta Q$). The enhanced $\Delta Q$ slows the tropospheric zonal circulation at the equator[15–17], which is referred to as the Walker Circulation. As a result, a westerly zonal-wind stress anomaly develops ($\Delta Q \rightarrow \tau_x$). In response to the wind stress anomaly, the zonal slope of the equatorial thermocline is reduced ($\tau_x \rightarrow dh/dx$), which is associated with an increase in upper-ocean heat content in the east and a drop in the west. As heat content increases, sea surface height (SSH) also rises[2,9] ($dh/dx \approx dSSH/dx$). Therefore, SSH can be used as a proxy for upper-ocean heat content[18]. Finally, through the so-called thermocline feedback the initial positive SST anomaly in the east is reinforced ($dSSH/dx \rightarrow dSST/dx$). The change in $\Delta Q$ drives the atmospheric circulation and is thus strongly involved in the positive ocean-atmosphere feedback that is summarized by the Bjerknes-feedback loop:

$$dSST/dx \rightarrow \Delta Q \rightarrow \tau_x \rightarrow dSSH/dx \rightarrow dSST/dx$$

The Bjerknes-feedback loop is certainly an important factor in the interannual equatorial Atlantic SST variability[1–6,9,11]. However, the physics underlying the Atlantic Niño remain controversial[4]. In particular, the relative roles of atmospheric and oceanic processes in the Atlantic Niño are under debate.

Here, by investigating the seasonality of the Atlantic Niño, we provide new insight into the role of atmospheric forcing for its interannual variability. We analyze satellite-derived estimates of SST[19] and SSH[20] in combination with $\Delta Q$ and wind stress ($\tau$) from multiple reanalysis systems for the period 1982–2015. It is shown that the seasonal migration of the Inter-Tropical Convergence Zone (ITCZ) modifies the background conditions over the equatorial Atlantic, which strongly influences the development of SST anomalies in this region. Hence, the seasonality of the Atlantic Niño is primarily set by the atmosphere.

## Results

### Seasonality of atmospheric diabatic heating and thermocline feedback.
The level of interannual equatorial Atlantic SST variability exhibits a marked seasonality with its maximum in boreal summer[1,2,4,11] (dashed curve Fig. 1c, d). Monthly standard deviations of the SST anomalies averaged over the Atl3 region (3°S–3°N, 0°–20°W) peak in June. The SSH variability (from the AVISO satellite altimetry, black curve in Fig. 1a) averaged over the full zonal extent of the equatorial Atlantic, a proxy for the vertical movement of the thermocline, exhibits a marked minimum in boreal spring followed by stronger variability during the remainder of the calendar year. The standard deviation of the basin-averaged vertical diabatic-heating gradient, $\Delta Q$, depending on data set, peaks 1–2 months earlier than the standard deviation

of the Atl3 SST and is very small during the remainder of the calendar year (Fig. 1b). The results do not fundamentally change when the variables are averaged over the Atl3 region (Supplementary Fig. 1). We note some spread among the different SSH and $\Delta Q$ reanalysis products.

Previous studies have shown the importance of the so-called thermocline feedback in the eastern equatorial Atlantic[1–6], which is one component of the Bjerknes-feedback loop: a deeper thermocline forces warmer SST. We note that the thermocline feedback is the dominant SST-anomaly growth mechanism in the eastern equatorial Pacific[1,2,12,13]. To explore the seasonality of the thermocline feedback in the equatorial Atlantic, for each calendar month we first calculate regressions of the Atl3-SST anomalies on the basin-averaged SSH variability, with SSH serving as a proxy for thermocline depth. The thermocline feedback exhibits two maxima of equal strength, one in June and one in November, which is observed in all data sets (Fig. 1c). The Atl3-SST anomalies exhibit interannual variability maxima in these two months too (dashed line in Fig. 1c, d), supporting an important role of the thermocline feedback in driving SST anomalies in the eastern equatorial Atlantic. However, the SST variability is much larger in June than in November, suggesting that SST variability in November is more strongly damped than in June. Based on averages over the Atl3 region, the thermocline feedback that here represents the impacts of local deepening and shoaling of the thermocline on SST variability in that region, displays a stronger peak in June (Supplementary Fig. 1 in the Supporting Information). Using instead of SSH the depths of the 20 °C ($Z_{20}$) and 23 °C isotherms ($Z_{23}$) in the calculations, as alternative definitions of thermocline depth, shows more spread but June and November–December peaks in thermocline feedback are still present in some data sets (Supplementary Fig. 2). In addition, there are negative anomalies in July using $Z_{23}$, suggesting an outcropping of the 23 °C isotherm in that month.

An analogous regression analysis of the Atl3 SST anomalies is performed on the diabatic-heating gradient $\Delta Q$ (Fig. 1d). The largest regressions are observed in boreal spring with peak values in May, except in the CFSR data set where the peak is observed one month later. In June, the interannual SST variability peaks and then the regressions attain minima a few months after the May–June maxima. Thus, the Atl3 SST anomalies appear to be most sensitive to $\Delta Q$ variability in boreal spring, which is the time when the maximum precipitation is close to the equator.

In order to compare the relative roles of the atmospheric diabatic heating and thermocline feedback for the seasonality of the Atlantic Niño, we estimate for each calendar month the proportion of the interannual SST variability that is explained by $\Delta Q$ and the thermocline proxies (Fig. 2). The $\Delta Q$ explains the largest variance during the development phase of the Atlantic Niño in boreal spring, with largest explained variances of the order of about 60% in May (Fig. 2a). On the other hand, the AVISO SSH anomalies averaged over the full extent zonal extent of the equatorial Atlantic as a proxy for the vertical movement of the thermocline account for about 80% of the SST variability in November (Fig. 2b). This peak corresponds to the so-called Atlantic Niño II observed in November–December[21], and shares similarities to the peak in ENSO variability occurring in boreal winter. A secondary peak of the variance explained by AVISO-SSH, amounting to about 45%, is observed in early boreal summer. Despite the large spread, all three thermocline proxies (SSH, $Z_{20}$ and $Z_{23}$) yield a major peak in the explained variance in November–December and smaller peaks between May and July (Fig. 2b–d), which implies that the thermocline feedback is stronger in November–December.

Next, the six SSH data sets and wind stress from the five reanalysis products are regressed on the Atl3 SST index in June

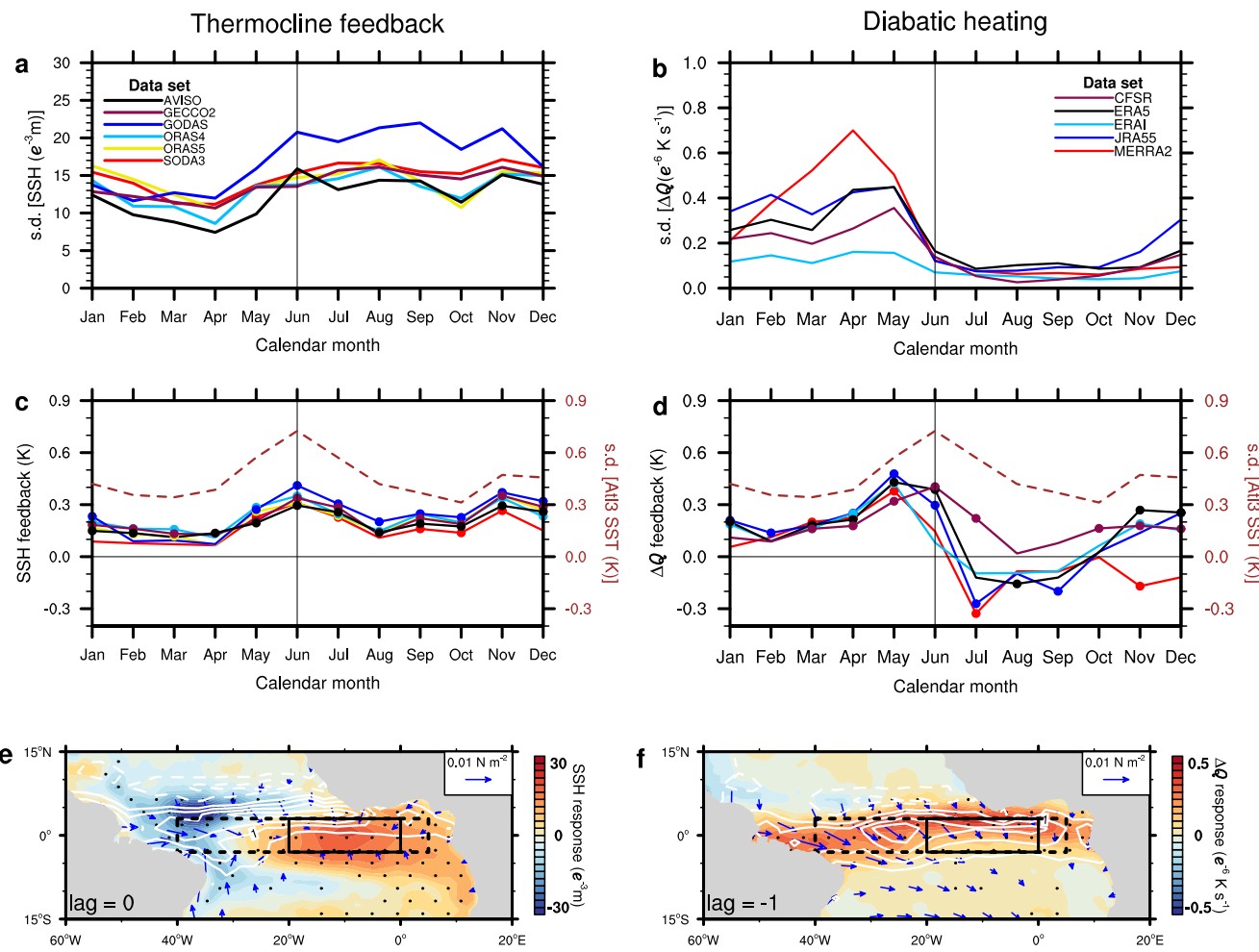

**Fig. 1 Seasonality of thermocline feedback and diabatic heating.** Standard deviations of seasonally stratified **a** sea surface height (SSH) and **b** diabatic heating gradient (ΔQ) averaged over the equatorial Atlantic region (3°N–3°S, 5°E–40°W; shown by dashed boxes in Fig. 1**e** and **f**) in multiple data sets. **c** (left scale) Thermocline feedback ($K$) calculated as Atl3 SST (that is, average in the region 3°N–3°S, 0°–20°W shown by the solid boxes in Fig. 1**e** and **f**) regressed on the normalized basin SSH index for each calendar month. **d** (left scale) Atl3 SST regressed on the normalized basin ΔQ index for each calendar month. In **c** and **d**, the right scale shows the seasonally stratified standard deviations of the Atl3 SST ($K$). Symbols on the lines denote statistical significance at the 95% confidence level. **e** The in-phase anomalies of SSH (color scale), wind stress (only statistically significant vectors are plotted) and precipitation (white contours, at interval of 0.5 mm day$^{-1}$) regressed on the normalized Atl3 SST in June. **f** The anomalies of ΔQ (color scale), wind stress (only statistically significant vectors are plotted) and precipitation (white contours, at interval of 0.5 mm day$^{-1}$) in May regressed on the normalized Atl3 SST index in June. In **e** and **f**, the SSH, ΔQ and wind stress are based on the ensemble-means of the different data products (see Methods); stippling denotes statistical significance at the 95% confidence level.

and ensemble-mean patterns shown (Fig. 1e). Anomalously warm Atl3 SST anomalies are associated with positive SSH (or thermocline depth) anomalies in the eastern and central basin and negative SSH anomalies in the west[1,2]. The pattern of anomalous SSH with its Kelvin/Rossby wave-like structure is consistent with the ensemble-mean wind stress regressions (arrows in Fig. 1e), exhibiting westerly anomalies over the western equatorial Atlantic. Precipitation anomalies[22] (white contours), a proxy for the vertically integrated diabatic-heating variability[23], associated with June-SST anomalies exhibit a dipolar structure with enhanced precipitation close to and at the equator and reduced precipitation further north, suggesting a southward migration of the ITCZ. Largest precipitation anomalies are observed over the western equatorial Atlantic. Overall, the in-phase patterns linked to the Atl3-SST variability in June are somewhat reminiscent of what is observed during El Niño events in the equatorial Pacific in December (Supplementary Fig. 3), except that the Atlantic precipitation anomalies are stronger north of the equator.

The reanalysis ΔQ and wind stress anomalies, as well as precipitation anomalies in May are regressed on the Atl3-SST index in the following June (Fig. 1f). Regions of positive anomalies in precipitation and diabatic heating largely coincide, supporting the notion that precipitation is a good proxy for diabatic heating over the equatorial Atlantic. One month prior to the SST anomalies in June, significant diabatic heating of the atmosphere is observed over the entire equatorial belt. The ΔQ is associated with wind-stress anomalies that are predominantly northwesterly west of 0°. Strongest wind-stress anomalies are observed in the very west where they are nearly westerly. Overall, the wind-stress anomalies imply a weakening of the prevailing southeasterly trade winds. The westerly wind-stress anomalies at the equator act to enhance the dynamical coupling between the ocean and the atmosphere[24,25]. Over the Pacific, the November patterns linked to the Niño3 index in the following December exhibit almost no diabatic heating over the Niño3 box (Supplementary Fig. 3). This is a major difference to the equatorial Atlantic exhibiting significant diabatic heating over

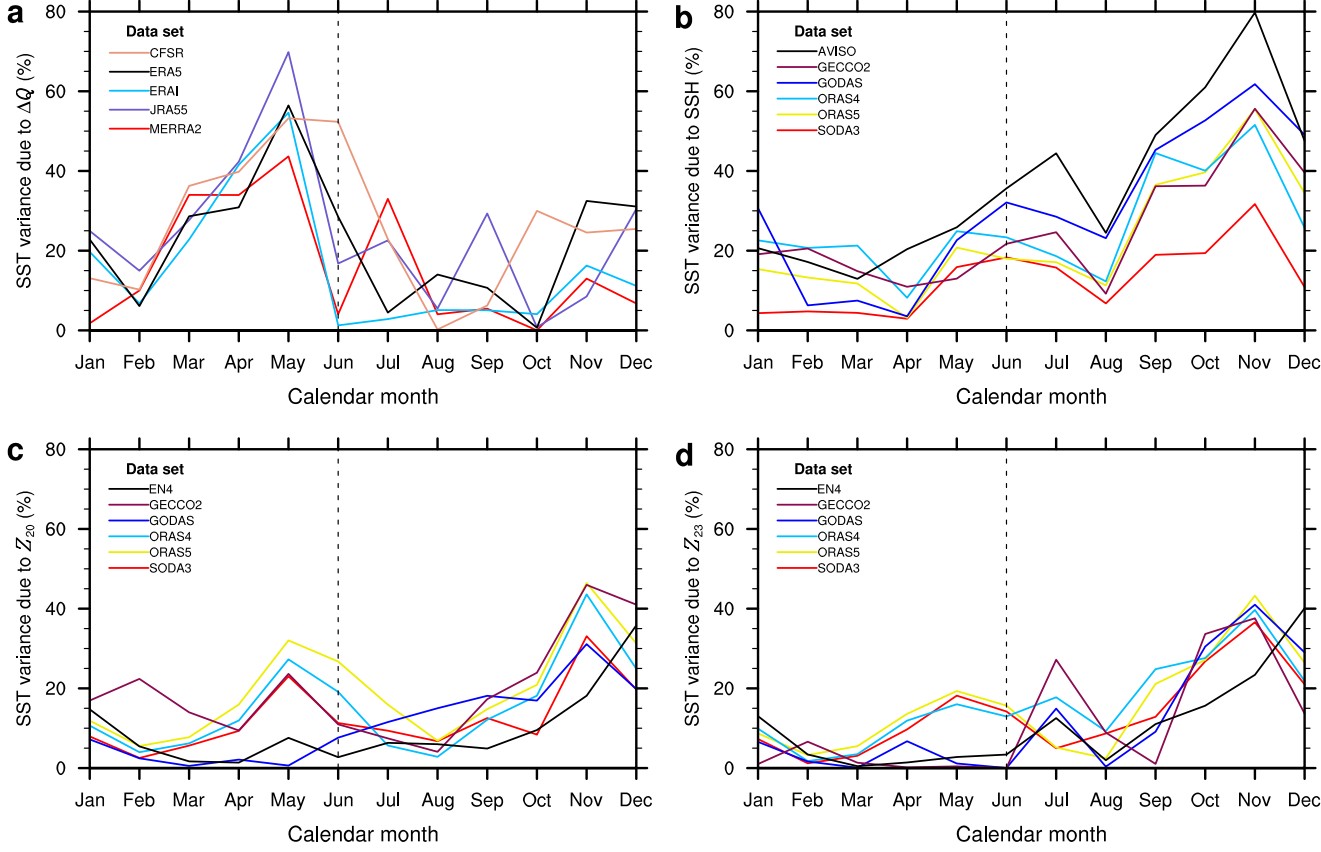

**Fig. 2 Sea surface temperature explained variances.** The explained variances were estimated from $r^2$ and expressed as percentages: where $r$ is the correlation between the seasonally stratified Atl3 SST (that is, average in the region 3°N–3°S, 0°–20°W) and **a** diabatic-heating gradient ($\Delta Q$), **b** sea surface height (SSH), **c** 20 °C isotherm depth ($Z_{20}$) and **d** 23 °C isotherm depth ($Z_{23}$) averaged over the equatorial Atlantic (3°N–3°S, 5°E–40°W).

the entire equatorial region (Fig. 1f). Finally, we note that a weakening of the Southern Hemisphere St. Helena subtropical anticyclone is suggested by the wind stress, which has been shown to influence the development of Atlantic and Benguela Niños[26–29].

**Meridional propagation of the atmospheric convection**. We hypothesize that the SST anomalies in the equatorial Atlantic are insufficient to keep the ITCZ close to the equator, which inhibits stronger and more persistent coupled ocean-atmosphere inter-actions in the form of the Bjerknes-feedback loop. The ITCZ generally coincides with the latitude of zero meridional wind stress ($\tau_y = 0$) and is identified here along 20°W at the western edge of the Atl3 region.

We depict the precipitation anomalies (along 20°W) regressed on the Atl3-SST anomalies in June as a function of calendar month and latitude (Fig. 3a). In Supplementary Fig. 4a, we show a similar analysis using outgoing longwave radiation (OLR), which is a good proxy of precipitation and diabatic heating[30]. The calendar months are expressed as time lags such that negative (positive) lags denote months prior to (after) June (lag = 0), when the peak in interannual SST variability is observed. Large precipitation anomalies are observed from lag = −2 to lag = +3. The precipitation anomalies, which are located slightly south of the climatological maximum rainfall, generally follow the northward migration of the climatological ITCZ[25]. Based on the wind stress data, we estimate the meridional displacements of the ITCZ at the 95% confidence level as two standard deviations around the mean

(±2σ, shown by the thin dashed curves in Fig. 3a and Supplementary Figs. 4 and 5).

There is a clear asymmetry in the positive precipitation anomalies at the equator linked to the June Atl3-SST variability, with much larger anomalies at negative than positive lags. Similarly, there are robust decreases in precipitation anomalies to the north of the ITCZ at negative lags. As the ITCZ propagates farther to the north at positive lags, precipitation anomalies become smaller at the equator. The northward propagation of the precipitation anomalies is closely reproduced along longitude 28° W, the longitude at which the ITCZ has been identified in previous studies[31,32], and using the basin-averaged data (Supplementary Fig. 5).

The precipitation anomalies averaged over the equatorial region (3°N–3°S, 5°E–40°W) linked to the June-SST anomalies are strongest at lag = −1, i.e., they lead the June-SST anomalies by one month but quickly diminish thereafter (Fig. 3a, b). This analysis suggests that mean atmospheric convection supports the growth of the SST anomalies at short negative lags, as already shown in Fig. 1d, f. However, the atmospheric response to the Atl3-SST anomalies at positive lags is small due to the lack of mean convection and a diabatic heating response at the equator. In contrast, SST anomalies lead precipitation anomalies in the Pacific Niño3 region, consistent with SST-forced equatorial precipitation variability (Fig. 4 and Supplementary Fig. 6b). This constitutes a fundamental difference in how the ocean and atmosphere interact in the equatorial cold-tongue regions of the Atlantic and Pacific Oceans.

Positive zonal-wind stress anomalies ($\tau_x$) over the western equatorial Atlantic (3°N–3°S, 20°W–40°W) that are linked to the

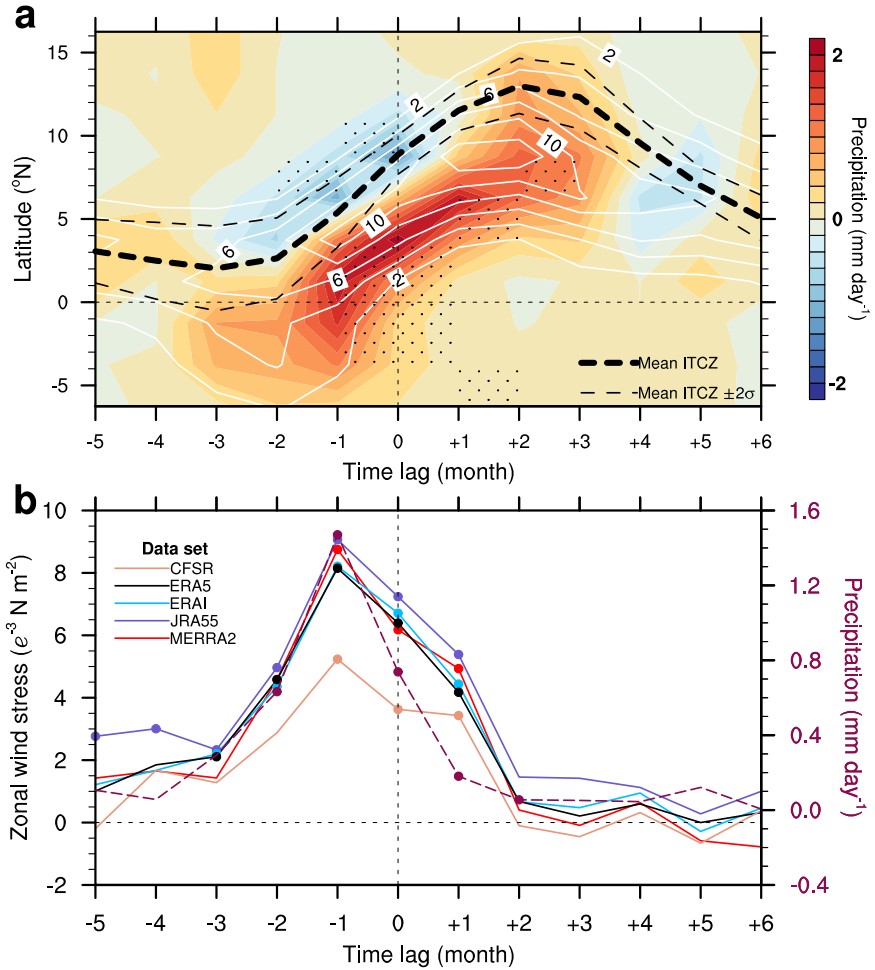

**Fig. 3 Related precipitation and zonal-wind stress variability. a** Seasonally stratified precipitation anomalies along 20°W regressed on the June (lag = 0) normalized Atl3 SST index (that is, average in the region 3°N–3°S, 0°–20°W). Stipples denote statistical significance at the 95% confidence level. The white contours show the climatological-mean precipitation, at interval of 2 mm day$^{-1}$. The dashed black curves denote the mean inter-tropical convergence zone (ITCZ) defined as the latitude of zero meridional wind stress (thick curve) and its two standard deviations represented by ±2σ (thin curves). **b** Right axis, dashed curve: precipitation anomalies averaged over the equatorial Atlantic region (3°N–3°S, 5°E–40°W) in all calendar months regressed on the normalized Atl3 SST index fixed in June (lag = 0). Left axis: zonal-wind stress averaged over the western equatorial Atlantic (3°N–3°S, 20°–40°W) in all calendar months regressed on the normalized Atl3 SST index fixed in June (lag = 0). Circular ticks in **b** denote statistical significance at the 95% confidence level.

Atl3-SST anomalies in June also peak at lag = −1 (Fig. 3b). Consistent with the gradient of diabatic heating ΔQ, the peak of the variability in precipitation, ITCZ position and $τ_x$ is observed prior to the peak in SST variability (Figs. 1b and Fig. 3; and Supplementary Figs. 6a and 7). The positive $τ_x$ at lag = −1 represents a slackening of the prevailing trade winds. This slackening is associated with the ITCZ crossing the equator, which tends to strengthen the zonal-wind component of the Bjerknes-feedback loop[2,24,33]. The ocean responds to the westerly wind stress anomaly by deepening the thermocline (Fig. 1e) and thus increasing the heat content in the eastern equatorial Atlantic, which leads to warmer SSTs in this region[1–5,11,34]. However, the SST anomalies in the east do not drive a strong atmospheric response along the equator due to a lack of diabatic heating after June.

**Ocean-mixed-layer heat budget**. To determine the processes that govern the growth and decay of the SST anomalies associated with a typical Atlantic Niño event (Fig. 5a), we calculate the ocean-mixed-layer heat budget (see "Methods"). This allows us to

directly compare the roles of surface heating, advective heat transport and entrainment associated with the Atlantic Niño. The time derivative of the mixed-layer temperature ($∂T/∂t$) peaks in May in all reanalysis-data sets and is accompanied by a sharp decline during the subsequent summer months (Fig. 5b–f)). Entrainment ($1/h[T − T_{-h}]w_e$), representing the tendencies in mixed-layer temperature due to vertical flows of water masses across the base of the mixed layer, is the dominant heating term associated with the peak SST anomalies in June, consistent with a previous modeling analysis[35]. Increased ($1/h[T − T_{-h}]w_e$) in that month implies either a reduction in upwelling or a deepening of the thermocline in the eastern basin (Fig. 1), corresponding to an increased heat content in that region, or both. The horizontal temperature advection term ($u∂T/∂x + v∂T/∂y$) also shows some enhancement in June, but horizontal advection is smaller than entrainment in all data sets. The tendencies due to the net surface heat flux ($Q_{net}/ρC_wh$), which drive the mixed-layer temperature anomalies in late boreal winter and especially in boreal spring[36], is the major damping term in June and July when the SST anomalies attain peak strength and are due to ocean dynamical processes.

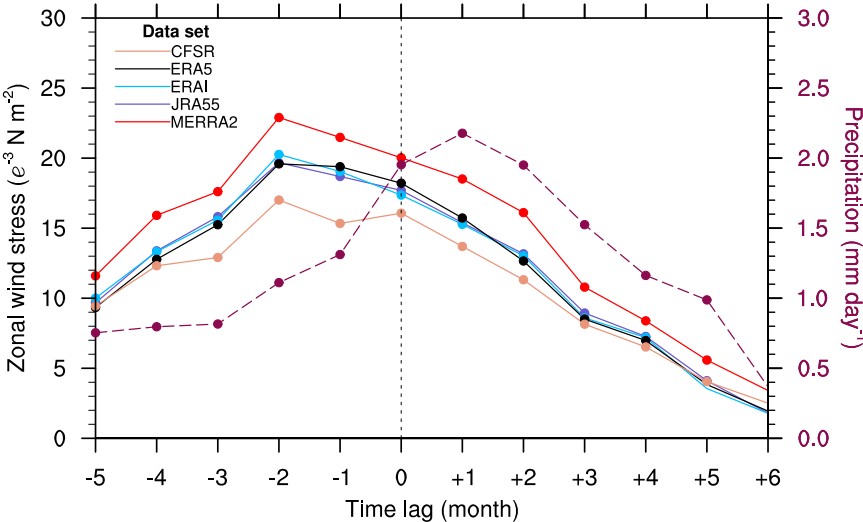

**Fig. 4 Pacific El Niño-type precipitation and wind stress variability.** Right axis, dashed curve: precipitation anomalies averaged over the Pacific Niño3 and Niño4 regions (5°N–5°S, 160°E–90°W) in all calendar months regressed on the normalized Niño3 SST index (that is, average in the region 5°N–5°S, 90°–150°W) fixed in December (lag = 0). Left axis: zonal-wind stress averaged over the Niño4 region (5°N–5°S, 160°E–150°W) in all months regressed on the normalized Niño3 SST index fixed in December (lag = 0). Circular ticks denote statistical significance at the 95% confidence level.

It must be stressed that the residual term ($\epsilon$) in the heat budget, representing both errors in the data and unconsidered or unresolved processes, such as mixing, in the heat budget analysis, is large in all five analyzed reanalysis products. The change in thermocline depth, for example, leads to a change in the mixed-layer temperature in the east by upwelling and vertical mixing[37], where the latter, while important, is not explicitly treated in our calculations and hidden in the residual term. In fact, the $\epsilon$ term is large during the months when the SST anomalies are strongest, suggesting that vertical mixing, by which subsurface-temperature anomalies are transported vertically in the upper layer of the ocean, could be an important contributor[37] to the tendency of the mixed-layer temperature. There also is a possibility that the entrainment term is underestimated since it was computed with monthly data[38].

The net surface heat flux ($Q_{net}$) constitutes a known source of observational uncertainty in the equatorial Atlantic[4,39,40], and the reanalysis systems (especially of the ocean) tend to overestimate $Q_{net}$ damping (Supplementary Fig. 8). Here, we try to reduce this contribution to the $\epsilon$ term by using a high-quality $Q_{net}$ data from the OAFLUX archive—derived from in situ measurements of the turbulent heat[41] and satellite-derived radiative[42] fluxes—in the heat budget calculations. Although the turbulent heat fluxes may be still biased[39,40], the spread of the $\partial T/\partial t$ terms in Fig. 5 is assumed to be largely reflecting differences in oceanic processes, as exemplified by the impacts of the ocean-mixed-layer depth seasonality (Supplementary Fig. 9) on $Q_{net}/\rho C_w h$ (Fig. 5). Nevertheless, there is a general indication that the $Q_{net}/\rho C_w h$ drives (damps) the SST anomalies in boreal spring (summer).

The heat budget analysis provides a link between atmospheric diabatic heating, winds, net heat flux and thermocline feedback in the Atlantic Niño region. The presence of deep atmospheric convection and diabatic heating in boreal spring, amplifies the atmosphere's sensitivity to equatorial SST anomalies[24] during this time of the year and supports changes in zonal-wind stress over the western equatorial Atlantic[1] (Figs. 1 and 3). The wind stress anomalies in the west, with a time delay of about a month, influences the SST[1,2,6] in the east by affecting the thermocline there. As the thermocline feedback strengthens and SST anomalies grow, the net heat flux switches sign and acts as a

damping (Fig. 5) because the ITCZ moves farther north of the equator during boreal summer (Fig. 3).

## Discussion

Observations and reanalysis products indicate that the seasonality of the Atlantic Niño is largely governed by the variability in atmospheric diabatic heating that is linked to the seasonal meridional migration of the ITCZ. The strongest diabatic-heating variability leads the strongest SST variability in June by about one month. Owing to the meridional migration of the ITCZ, strong SST variability at the equator is limited to a relatively short period of the calendar year—the late boreal spring and early summer (May–July). After June, the atmosphere responds only weakly to equatorial SST anomalies. Atmospheric model experiments support a generally weak role of equatorial Atlantic SST in forcing precipitation and zonal-wind stress[43]. Such a picture is in part inconsistent with the Bjerknes-feedback loop, which does involve a strong feedback from the ocean to the atmosphere in the form of SST-forced atmospheric variability, as observed over the equatorial Pacific[12,13] and shown here in a companion analysis. This difference in ocean-atmosphere interaction between the Atlantic Niños and the Pacific El Niños may explain why SST variability is weaker[1,2,5,34] and SST predictability lower[44,45] in the equatorial Atlantic relative to that in the equatorial Pacific. These findings raise questions about the applicability of the standard Bjerknes-feedback loop as the major explanation for the mechanism underlying the Atlantic Niño.

There appears to be a subtle balance between thermodynamic and dynamical processes determining the SST variability in the equatorial Atlantic. The small oceanic feedback on the atmosphere may explain why thermodynamic processes, which, for example, only are represented in the so-called slab ocean coupled models, can force significant interannual SST variability in the eastern equatorial Atlantic[36,46]. Biased ocean dynamics in fully coupled general circulation models (e.g., the lack of a sufficiently strong cold tongue in the equatorial Atlantic[47]), however, may lead to an overestimation of the impacts of thermodynamic processes on the SST variability[36].

While the Bjerknes-feedback loop operates in the zonal direction, the atmospheric deep convection influencing the

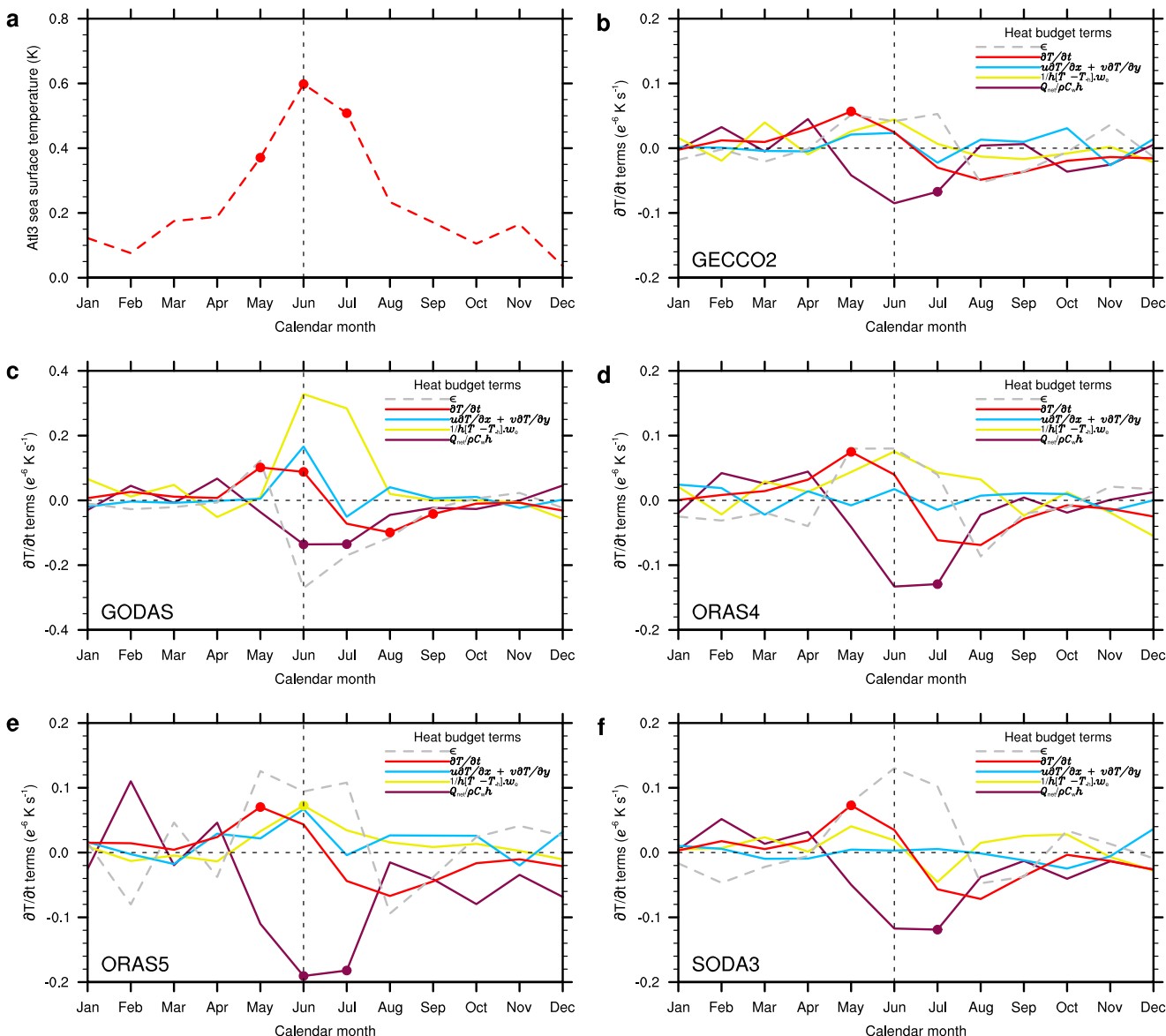

**Fig. 5 Composites of the Atlantic Niño heat budget. a** Composite average of 10 Atlantic Niño events, defined as Atl3 SST index (that is, average in the region 3°N–3°S, 0°–20°W) greater than or equal to 0.5 K persisting for two overlapping three-month seasons, for the period 1984–2009 (see "Methods"). **b**–**f** Composite average of the ocean-mixed-layer heat budget terms for the 10 Atlantic Niño events in the different ocean reanalysis data sets indicated on the bottom-left corner of each panel. The heat budget terms shown are the time derivative of the temperature ($\partial T/\partial t$) and the tendencies due to the surface net heat flux ($Q_{net}/\rho C_w h$), advective transport ($u\partial T/\partial x + v\partial T/\partial y$), entrainment ($1/h[T - T_{-h}]w_e$) and unresolved processes ($\epsilon$). Dots denote statistical significance at the 95% confidence level. For all terms, positive (negative) values mean heating (cooling) of the ocean-mixed-layer. Note that the vertical scale for GODAS data set is twice as large.

Atlantic Niño propagates in the meridional direction, northward during the boreal spring and summer. Strong ocean-atmosphere coupling in the equatorial Atlantic appears to be more short-lived than previously thought and is associated with the ITCZ and diabatic-heating belt crossing the equator in May. The coupling at the equator decays rapidly during the boreal summer due to the disappearance of the atmospheric deep convection as the ITCZ propagates far north of the equator. This implies that the window of climate predictability in the equatorial Atlantic could be small and largely confined to the robust coupling phase in boreal spring and, due to the persistence of the SST anomalies[45] in the subsequent summer. Consequently, accurate representation of the state of the atmosphere and the upper ocean during the boreal spring could prove vital for improved seasonal climate predictions over the tropical Atlantic.

Remote forcing signals in the ocean and atmosphere may represent additional sources of predictability. Previous studies suggest some roles of remote forcing from the Benguela region[26,29], tropical Pacific[48–51], the extratropical South Atlantic[29] and North Atlantic[52], meridional temperature advection from the tropical North Atlantic[53], the Atlantic meridional mode —including the impact of the associated cross-equatorial SST gradient on the ITCZ[17,32,54]. However, it remains unclear how the remote signals interact with the meridional propagation of the deep convection in the atmosphere over the tropical Atlantic and with the Atlantic Niño. As the seasonal meridional ITCZ migration is robustly projected to be delayed over the tropical Atlantic region in the future[55,56], it is necessary to better understand the dynamics of interannual climate variability in the equatorial Atlantic and its interaction with the ITCZ.

## Methods

**Data**. Continuous global satellite observations of SST[19] and precipitation[22] over the ocean date to the early 1980s and that of SSH[20] to the early 1990s (see Supplementary Table 1). $Q_{net}$ has been estimated from a combination of in situ measurements of turbulent fluxes[41] and satellite-derived radiative fluxes[42]. Direct measurements of atmospheric diabatic heating are not available. However, the current generation of atmospheric data assimilation systems routinely provide diabatic-heating fields[57–61] (Supplementary Table 2). Here, the vertical gradient of the diabatic heating $\Delta Q$ is defined as the difference between the mid-troposphere (400 hPa) and the surface layer (925 hPa): $\Delta Q = Q_{400hPa} - Q_{925hPa}$.

We represent the atmospheric variability using the $\Delta Q$, precipitation, $\tau$, $Q_{net}$ and OLR[62] and estimate the thermocline feedback using SSH data based on the satellite-derived "absolute dynamic topography" from AVISO[20] and multiple ocean reanalysis systems[63–67] (Supplementary Table 3). We also estimated the isotherm definitions of the thermocline using the $Z_{20}$ and $Z_{23}$ from these reanalysis data sets and the in situ derived EN4[68]. These choices, supported by the full ocean-mixed-layer heat budget, allow us to show the relative roles of the ocean and atmosphere for seasonality of the Atlantic Niño.

The analysis is based on the satellite era 1982–2015 with generally improved observations over the ocean compared to the previous decades. Nonetheless, uncertainties remain, especially in the reanalyses[4], and these are addressed by using multiple reanalysis data sets of the ocean and atmosphere.

**Calculation of the ocean-mixed-layer heat budget**. The time derivative of the temperature averaged over the ocean-mixed-layer is governed by the tendencies due to horizontal ocean currents, vertical flow of mass across the base of the ocean-mixed-layer or entrainment and the net surface heat flux, as well as a residual term:

$$\frac{\partial \langle T \rangle}{\partial t} = \underbrace{\left\langle u\frac{\partial T}{\partial x} + v\frac{\partial T}{\partial y} \right\rangle}_{horizontal\,advection} + \underbrace{\frac{1}{h}[\langle T \rangle - T_{-h}]w_e}_{entrainment} + \underbrace{\frac{Q_{net}}{\rho C_w h}}_{heat\,flux} + \epsilon \quad (1)$$

$T$ is the ocean-mixed-layer temperature and $u$ and $v$ are the zonal and meridional ocean current velocities, respectively. $\rho$ and $C_w$ are constants representing the sea water density ($\rho = 10^3\,\mathrm{kg\,m^{-3}}$) and and specific heat capacity ($C_w = 4 \times 10^3\,\mathrm{J^{-1}kg^{-1}K}$), $h$ is the interannual-varying monthly mean mixed-layer depth, $T_{-h}$ is the temperature at the base of the mixed-layer, $w_e$ is the entrainment velocity and $Q_{net}$ the net surface heat flux. $\epsilon$ is the residual term that represents the sum of unresolved physical processes (such as mixing and high-frequency variability that is not resolved by the monthly-mean time series used here) and errors arising from averages over the Atl3 region. $T$, $u$, and $v$ are based on the vertical averages over the depth of the ocean-mixed-layer ($h$):

$$\langle \bullet \rangle = \frac{1}{h}\int_0^h \bullet\,dz \quad (2)$$

$h$ is not available from EN4, GECCO2, and ORAS4 data sets and was estimated as the depth at which ocean temperatures are 0.5 K lower than those at the surface. Similarly, the $h$ determined by temperature criterion is used for SODA3, which provides multiple definitions for $h$. $w_e$ is the entrainment velocity at the base of the mixed-layer and calculated as follows:

$$w_e = w_{-h} + \frac{\partial h}{\partial t} + \mathbf{U}_{-h}.\nabla h \quad (3)$$

$w_{-h}$ and $\mathbf{U}_{-h}$ are the vertical velocity and horizontal current vector at the base of the mixed-layer, respectively. $w$ is not available from ORAS4 and ORAS5 archives and was calculated from the divergence of the horizontal current velocities and then vertically integrated at all ocean levels from the surface ($z = 0$) to the bottom ($z = h_b$):

$$w = \int_0^{h_b} \frac{\partial w}{\partial z}\,dz = \int_0^{h_b} -\left(\frac{\partial u}{\partial x} + \frac{\partial v}{\partial y}\right)dz \quad (4)$$

The individual heat budget terms used to construct the composites in Fig. 5 are based on area averages over the Atl3 region.

**Statistical analysis**. Each data set was seasonally stratified and then the least-squares linear trend and long-term term mean removed to create monthly anomalies. We conducted least-squares regression analysis and composite analysis and tested for statistical significance using a two-tailed Student $t$-test with the 95% confidence levels marked. The regressions were calculated for the period 1982–2015 (1993–2015 for AVISO and 1982–2010 for CFSR) first using the individual data sets. Then, the ensemble-mean maps of the SSH, $\Delta Q$, and $\tau$ were calculated (Fig. 1e, f and Supplementary Fig. 3). Autocorrelation was accounted for in statistical significance tests for the regression coefficients by adjusting the number of degrees of freedom of the time series pairs as follows[69]:

$$N^* = N\frac{(1 - r_1 r_2)}{(1 + r_1 r_2)} \quad (5)$$

where $N$ is the length of the time series; $r_1$ and $r_2$ are the lag-1 autocorrelation coefficients of the time series, respectively; and $N^*$ is the adjusted number of degree of freedom used for determining the statistical significance.

The composite evolution of the ocean mixed heat budget in the Atl3 region was constructed to show the processes that govern typical Atlantic Niño events. The composite was based on 10 events (1984, 1987, 1988, 1991, 1993, 1995, 1996, 1998, 1999, 2008) between 1984 and 2009 during which $Q_{net}$ the observational OAFLUX data set is available. These events were identified in a recent study[70] based on the persistence of Atl3-averaged SST anomaly of equal to or greater 0.5 K for two consecutive overlapping 3-month seasons.

## Data availability

The observational[19,20,22,41,62,68], atmospheric reanalysis[57–61] and ocean reanalysis[63–67] data sets used are publicly available at the sources indicated in Supplementary Tables 1, 2 and 3, respectively.

## Code availability

Codes for the data analysis and preparation of the figures are freely available at https://github.com/hnnamchi/dheatingNatComm.

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

## Acknowledgements

This study was supported by the BANINO project of the German Ministry of Education and Research (BMBF). H.C.N. was funded by the Alexander von Humboldt Postdoctoral Fellowship. N.S.K. acknowledges support from EU H2020 program (STERCP, grant 648982; TRIATLAS, grant 817578). M.L. acknowledges support from JPI Climate & JPI Oceans (ROADMAP, grant 01LP2002C). The authors thank Joke Lübbecke and Ping Chang for informal reviews of the initial draft.

## Author contributions

H.C.N. and M.L. developed the concept and led the writing of the manuscript. H.C.N. conducted the analysis. N.S.K., J.K., and I.R. contributed to discussion of the mechanisms and improvement of the manuscript.

## Funding

## Competing interests

The authors declare no competing interests.
