## [Peer Review File · Nature Communications]

REVIEWER COMMENTS, first round:

Reviewer #1 (Remarks to the Author):

Review of manuscript "Origin of the seasonality of the Atlantic Niño" submitted to Nature Communications by H. Nnamchi, M. Latif, N.S. Keenlyside, J. Kjellsson, and I. Richter

The manuscript by Nnamchi et al investigates a mechanism influencing the zonal model of the tropical Atlantic that has not yet been investigated, is compelling and will capture the interest of researchers in the field. It presents an interesting and thorough study of influences on the zonal mode derived from reanalysis and satellite derived observations and deserves to be published in Nature Communications. The analysis is carefully constructed and well carried out in most parts, some of the writing could be more precise, and I would appreciate re-wording. I recommend minor revisions.

General comments:

- 1) In order to compare the importance of the SST variability directly explained by diabatic heating should be compared to that explained by the components of the Bjerknes Feedback loop.
- 2) The analysis of Fig 4 seems to be based only on one data set (2 references being cited in line 181), if that is the case, please repeat this analysis with several datasets to convince the reader. There are large uncertainties in the turbulent heat fluxes (as shown in Nogueira Neto et al. (2018)), which should be taken into consideration.

Specific comments:

- Lines 50-51, the authors mention adjustment of the vertical gradient of atmospheric heating is a "complex interplay between dynamics and thermodynamics across the equatorial belt" - this statement is extremely vague. Especially since ΔQ is the main target of this study an explanation is needed. This part of the loop is often omitted, so it truly does deserve a couple more sentences. I was under the impression that the westerly zonal wind stress anomaly is seen as a direct result of the change of the zonal SST gradient.
- Lines 71-72, I do not see how the seasonality being set by the atmosphere is inconsistent with the Bjerknes Feedback. It could be that there are specific months in which the atmosphere is conducive to the Bjerknes Feedback loop which would set the seasonality, but does not mean that the Bjerknes Feedback is not at all active. Since ΔQ leads to wind stress changes in the western tropical Atlantic, could it not trigger the Bjerknes Feedback? This seems to be the interpretation in Keenlyside and Latif 2007 Figure 2. While the correlation is largest with wind stress leading, it does persist with SST leading (and you have accounted for auto-correlation), so I do not see how a coupled feedback can be excluded (see lines 223-227). Maybe it is better to frame it as the Bjerknes Feedback is more short-lived than previously thought?
- Lines 144-145, that sentence is a bit obscure - is the location chosen because it is deemed to be at the center of the basin? Has the analysis been repeated with different longitudes, such as closer to ATL3 or a wester Atlantic longitude? I do not understand what the reason is to use the center of the basin since the ITCZ slopes. What if the slope was used? Or two locations? The whole rest of the ITCZ surely cannot be influenced only by the land winds? I find these two sentences a little confusing (lines 144-146).
- Line 185: Figure 5d is cited, but shows wind stress, should this be Figure 5c?
- Line 220: should this be dynamics?

Nogueira Neto, A. V., Giordani, H., Caniaux, G., & Araujo, M. (2018). Seasonal and interannual mixed layer heat budget variability in the western tropical Atlantic from Argo floats (2007–2012). *Journal of Geophysical Research: Oceans*, 123, 5298–5322.
<https://doi.org/10.1029/2017JC013436>

Keenlyside, Noel S., and Mojib Latif. "Understanding equatorial Atlantic interannual variability." *Journal of climate* 20.1 (2007): 131-142.

Reviewer #2 (Remarks to the Author):

The paper suggests a novel change in our view of tropical Atlantic interannual climate variability. The very knowledgeable authors point out that previous studies assume that Bjerknes feedback dominates as it does with ENSO in the Pacific, which I confirmed with a brief survey of previous papers. Here I provide two comments made with the goal of strengthening their approach.

Evidence for the importance of diabatic heating seems to hinge on Figures 1c and 1d. In the former SST is regressed on SSH to represent the role of thermocline fluctuations. I find supplementary Fig. 1c more useful than Fig. 1c since S. Fig.1c directly compares SST and the movement of the thermocline. Incidentally, SSH in SODA3.3.1 is a diagnostic variable that needs to be corrected to remove the global average. That's why SODA3.3.1 SSH is so noisy compared to the other products compared in Fig. 1c, but SODA3.3.1 thermocline depth is not noisy compared to SST in S. Fig. 1c. Correcting SODA3.3.1 SSH to remove its global average should also address the comments in Lines 87-88 and 95-96 and explain this inconsistency between the main figure and supplementary figure.

While intriguing, the authors know that distinguishing coupled mechanisms based on lag covariances can be misleading due to cross-covariances among variables like winds and heating, and due to the presence of nonlinearities. More convincing for me would be to compute the full ocean mixed layer heat budget for ATL3. Diagnosing that mixed layer heat budget would allow the direct comparison of local heating, advective heat transport, and entrainment, all in the same units. I know several of the authors have done similar calculations in their previous studies. Diagnosing the mixed layer heat budget would also allow a better determination of the link to the seasonal variations of winds and surface heating. This study uses June as the month to focus on due to its high variability (L.82). But ATL3 SST is coldest, and thus the Bjerknes feedback is presumably strongest in August. A full heat budget can examine the monthly dependence of the feedback.

My other concern has to do with the how the authors are handling trends in the data. An examination of ATL3 SST and SSH shows that both have large warming/rising trends during 1982-2015 (<https://www.aoml.noaa.gov/phod/regsatprod/atl3/index.php>). For example, SST has warmed by 1C over this time period, while Fig. 1c shows a June SST standard deviation of ~0.7C. If they have not already done so (I didn't see discussion of this), I recommend removing trends before looking at the causes of interannual variability.

-- Jim Carton

Response to the specific comments and suggestions from Reviewer #1:

We very much appreciate your constructive and helpful comments. Your comments were invaluable in revising and improving our manuscript. Our response to each comment is written in blue italic. We have indicated where the changes occur in the present version of the manuscript in the specific responses below.

Reviewer #1 (Remarks to the Author):

Review of manuscript “Origin of the seasonality of the Atlantic Niño“ submitted to Nature Communications by H. Nnamchi, M. Latif, N.S. Keenlyside, J. Kjellsson, and I. Richter

The manuscript by Nnamchi et al investigates a mechanism influencing the zonal model of the tropical Atlantic that has not yet been investigated, is compelling and will capture the interest of researchers in the field. It presents an interesting and thorough study of influences on the zonal mode derived from reanalysis and satellite derived observations and deserves to be published in Nature Communications. The analysis is carefully constructed and well carried out in most parts, some of the writing could be more precise, and I would appreciate re-wording. I recommend minor revisions.

In the revised manuscript, we improved the writing and overall clarity. We have edited the manuscript to address your comments and for consistency with the additional analysis conducted.

We have performed the additional analysis you suggested. The SST explained variance due to the thermocline proxies (SSH, z20 and z23) and ΔQ variability is shown in Fig. 2 of this revised manuscript. We have also included the Q_{net} from multiple reanalysis of the ocean and atmosphere (Fig. S8).

Please see our responses to specific comments below.

General comments:

1) In order to compare the importance of the SST variability directly explained by diabatic heating should be compared to that explained by the components of the Bjerknes Feedback loop.

We have now calculated the the proportion of the SST variability explained by the diabatic heating and the thermocline proxies (SSH, Z_{20} and Z_{23}). The explained variance (Fig. 2) is discussed in this revised manuscript. (lines: 105-116).

This analysis clearly shows that ΔQ explains maximum variance during the development phase of the Atlantic Niño in boreal spring. On the other hand, the maximum explained variance by the thermocline proxies occur in November-December and only a smaller peak in May-July. These findings support our conclusion of a key role for diabatic heating variability which peaks in spring and related atmospheric variability for the development of the Atl3 primary peak in June.

2) The analysis of Fig 4 seems to be based only on one data set (2 references being cited in line 181), if that is the case, please repeat this analysis with several datasets to convince the reader. There are large uncertainties in the turbulent heat fluxes (as shown in Nogueira Neto et al. (2018)), which should be taken into consideration.

In this revised manuscript, we have analyzed the surface heat flux (Q_{net}) from multiple reanalysis data sets of the ocean (Fig. S8a) and atmosphere (Fig. S8b). Figure S8 shows a large spread of Q_{net} across the data sets. The spread is generally larger for the ocean reanalysis compared to the atmospheric reanalysis. Both atmospheric and oceanic reanalysis data sets generally tend to overestimate the (OAFLUX) observational Q_{net} .

These changes are reflected in the discussions of the error term in the mixed layer heat budget and the possible contributions of Q_{net} biases to the error term. (lines: 203-220).

Specific comments:

- Lines 50-51, the authors mention adjustment of the vertical gradient of atmospheric heating is a “complex interplay between dynamics and thermodynamics across the equatorial belt” - this statement is extremely vague. Especially since ΔQ is the main target of this study an explanation

is needed. This part of the loop is often omitted, so it truly does deserve a couple more sentences. I was under the impression that the westerly zonal wind stress anomaly is seen as a direct result of the change of the zonal SST gradient.

Based on suggestions, we have made edits to remove the unclear expressions and improved the overall clarity of the paragraph referred to. The link between SST anomalies, the vertical gradient of diabatic heating ΔQ , wind stress and SSH has been reformulated in more concise language. (lines: 43-57).

The link between ΔQ and the surface variability is further discussed in terms of the mixed-layer heat budget. (lines: 222-231).

- Lines 71-72, I do not see how the seasonality being set by the atmosphere is inconsistent with the Bjerknes Feedback. It could be that there are specific months in which the atmosphere is conducive to the Bjerknes Feedback loop which would set the seasonality, but does not mean that the Bjerknes Feedback is not at all active. Since ΔQ leads to wind stress changes in the western tropical Atlantic, could it not trigger the Bjerknes Feedback? This seems to be the interpretation in Keenlyside and Latif 2007 Figure 2. While the correlation is largest with wind stress leading, it does persist with SST leading (and you have accounted for auto-correlation), so I do not see how a coupled feedback can be excluded (see lines 223-227). Maybe it is better to frame it as the Bjerknes Feedback is more short-lived than previously thought?

Based on this suggestion we have removed “which is inconsistent with the Bjerknes mechanism” referred to, simply stating that the seasonality is primarily set by the atmosphere in this revised manuscript. (lines: 66-67).

The key message of our work is that ocean-atmosphere atmosphere coupling is short-lived, being strong between late spring and early summer. Diabatic heating (thermocline) plays stronger roles in spring (summer). The mixed-layer heat budget introduced in this revised version also supports strong roles for diabatic heating in spring (as seen in the Q_{net} tendency term) and thermocline feedback in summer (entrainment, horizontal advection). (Fig. 5). (lines: 188-231).

- Lines 144-145, that sentence is a bit obscure - is the location chosen because it is deemed to be at the center of the basin? Has the analysis been repeated with different longitudes, such as closer to ATL3 or a wester Atlantic longitude? I do not understand what the reason is to use the center of the basin since the ITCZ slopes. What if the slope was used? Or two locations? The whole rest of the ITCZ surely cannot be influenced only by the land winds? I find these two sentences a little confusing (lines 144-146).

This comment refers to discussions of the possible impacts of the continents on the ITCZ in the previous submission. We have now addressed this comment by repeating the analysis using two longitudes: 20°W on the edge of the Atlantic Niño region (Fig. 3a) and 28W (Fig. S5a) and by using the basin averaged data (Fig. S5b).

All three time-latitude plots show similar a northward propagation of the precipitation anomalies. Furthermore, they all show maximum precipitation anomalies along the equator in May and maximum precipitation to the north of equator in summer. We have also included a similar analysis of the OLR (Fig. S4) to provide additional support for the discussion of meridional propagation of atmospheric convection.

- Line 185: Figure 5d is cited, but shows wind stress, should this be Figure 5c?

This comment refers to the discussion of Q_{net} regression on SST which has now been replaced with the mixed-layer heat budget. The consistency of all figure discussions are checked in this revision.

- Line 220: should this be dynamics?

This sentence implies that coupled models with biased ocean dynamics (e.g., lack of a sufficiently strong cold tongue) may overestimate the impacts of surface heat fluxes (i.e., thermodynamic processes) on the SST variability. It has been edited to improve clarity. (lines: 253-255).

Suggested References:

Thank you for the suggestion of new referees, we discussed them in the revised manuscript.

Keenlyside, N. S., and M. Latif, 2007: Understanding Equatorial Atlantic Interannual Variability. *J. Climate*, **20**, 131–142, <https://doi.org/10.1175/JCLI3992.1>.

Nogueira Neto, A. V., Giordani, H., Caniaux, G., & Araujo, M. (2018). Seasonal and interannual mixed layer heat budget variability in the western tropical Atlantic from Argo floats (2007–2012). *Journal of Geophysical Research: Oceans*, **123**, 5298– 5322. <https://doi.org/10.1029/2017JC013436>

Response to the specific comments and suggestions from Reviewer #2:

We very much appreciate your constructive and helpful comments. Your comments were invaluable in revising and improving our manuscript. Our response to each comment is written in blue italic. We have indicated where the changes occur in the present version of the manuscript in the specific responses below.

Please see our responses to specific comments below.

Reviewer #2 (Remarks to the Author)

The paper suggests a novel change in our view of tropical Atlantic interannual climate variability. The very knowledgeable authors point out that previous studies assume that Bjerknes feedback dominates as it does with ENSO in the Pacific, which I confirmed with a brief survey of previous papers. Here I provide two comments made with the goal of strengthening their approach.

Evidence for the importance of diabatic heating seems to hinge on Figures 1c and 1d. In the former SST is regressed on SSH to represent the role of thermocline fluctuations. I find supplementary Fig. 1c more useful than Fig. 1c since S. Fig.1c directly compares SST and the movement of the thermocline.

In this revised manuscript, the basin-averaged SSH is now shown in Fig. 1c and used to discuss the movements of the thermocline in the main text. The the averages in the Atl3 region is moved to the Supplementary Information and used to discuss the localized thermocline variability in that region.

Incidentally, SSH in SODA3.3.1 is a diagnostic variable that needs to be corrected to remove the global average. That's why SODA3.3.1 SSH is so noisy compared to the other products compared in Fig. 1c, but SODA3.3.1 thermocline depth is not noisy compared to SST in S. Fig. 1c.

Correcting SODA3.3.1 SSH to remove its global average should also address the comments in Lines 87-88 and 95-96 and explain this inconsistency between the main figure and supplementary figure.

We have moved the analysis to a newer version of the SODA reanalysis (SODA v3.4.2) in this revised manuscript.

While intriguing, the authors know that distinguishing coupled mechanisms based on lag covariances can be misleading due to cross-covariances among variables like winds and heating, and due to the presence of nonlinearities. More convincing for me would be to compute the full ocean mixed layer heat budget for ATL3. Diagnosing that mixed layer heat budget would allow the direct comparison of local heating, advective heat transport, and entrainment, all in the same units. I know several of the authors have done similar calculations in their previous studies. Diagnosing the mixed layer heat budget would also allow a better determination of the link to the seasonal variations of winds and surface heating. This study uses June as the month to focus on due to its high variability (L.82). But ATL3 SST is coldest, and thus the Bjerknes feedback is presumably strongest in August. A full heat budget can examine the monthly dependence of the feedback.

As suggested we have now performed ocean mixed layer heat budget (Fig. 5) and introduced a new section to discuss the result. (lines: 188-231).

The heat budget analysis shows entrainment as the major driving term during the SST peak in June. Zonal advection tendency also peaks in June but is generally of smaller magnitude compared to entrainment in all reanalysis data sets. The Q_{net} tendency drives (damps) the SST anomalies in spring (summer).

Thus, the heat budget analysis supports strong roles for diabatic heating in spring (through the Q_{net} tendency) and thermocline feedback in summer (as seen in entrainment and horizontal advection). We have improved discussions of the link between atmospheric variability (diabatic heating, wind stress) and Q_{net} tendency in spring as well as the transition to ocean-driven variability in summer.

The ocean mixed heat budget is based on a composite of 10 Atlantic Nino events identified in a recent study (Vallès-Casanova et al., 2020, ref. 70) based on the persistence of Atl3-averaged SST anomaly of equal to or greater 0.5 K for two consecutive overlapping three-month seasons. Nonetheless, the SST composite peaks in June (Fig. 5a) consistent with the SST seasonality (Fig. 1c,d).

The analysis in Fig. 3 is focused on June to demonstrate the seasonal sequence of the maximum variability in the atmosphere and ocean at the equator.

The analyses of thermocline feedback are based on variables for the calendar months (e.g., Figs 1, 2). Using this month-by-month analysis, the thermocline feedback is generally strong throughout the year with peaks in boreal summer and winter (Fig. 1).

My other concern has to do with the how the authors are handling trends in the data. An examination of ATL3 SST and SSH shows that both have large warming/rising trends during 1982-2015 (<https://www.aoml.noaa.gov/phod/regsatprod/atl3/index.php>). For example, SST has warmed by 1C over this time period, while Fig. 1c shows a June SST standard deviation of ~0.7C. If they have not already done so (I didn't see discussion of this), I recommend removing trends before looking at the causes of interannual variability.

We removed linear trends from all data sets analyzed. This is now stated in the Methods section. (lines: 452-453).

REVIEWER COMMENTS, second round:
REVIEWER COMMENTS

Reviewer #1 (Remarks to the Author):

The revision has greatly improved the manuscript, and the addition of several datasets has strengthened the authors point.

I will note that I had some trouble following lines 222-231 which are fundamental to understanding the relationship between the heat budget Q_{net} and ΔQ (and thereby connection the heat budget to the mechanism described in this manuscript). Please consider rewording / reordering especially the sentence in lines 223-226.

This sentence explains the relationship between Q_{net} and ΔQ prior to the SST anomaly, do I assume correctly that after the SST anomaly Q_{net} is dominated by the warm anomaly rather than wind stress changes (and therefore ΔQ)? I assume this because there is predominantly positive correlation of ΔQ proxy precip with SST anomalies as shown in Fig 3b, but the Q_{net} becomes negative immediately as $dSST/dt$ peaks. If so please consider adding a sentence.

L 120 "are associated"

L 158-159 - does this not contradict your earlier statement that the SST anomalies are not high enough to keep the ITCZ on the equator? Or do we assume reverse causality?

L 238 I suggest: "After June, the atmosphere responds only weakly to SST anomalies."

Reviewer #2 (Remarks to the Author):

In my original review of "Origin of the seasonality of the Atlantic Niño"

In my review I made three requests, 1) to redo the material presented in Fig1c,d to more directly relate to variations in the thermocline, 2) I requested that they actually compute the mixed layer heat budget rather than draw inferences from covariances among variables, and 3) I asked them to clarify how they handled trends in the observational data as these can impact the results. These last two issues relate to statistical confidence. With these changes, and the changes made in response to the first reviewer I am happy to recommend publication.

The revised manuscript has addressed all three of these concerns. The authors and I also had a separate exchange regarding their inclusion of a product that my group produces and they, kindly, upgraded the product that they use (which provides more consistent results).

-- Jim Carton

Response to the specific comments and suggestions from Reviewer #1:

We very much appreciate your constructive and helpful comments. Your comments were invaluable in revising and improving our manuscript. Our response to each comment is written in blue italic. We have indicated all changes in the present version of the manuscript using tracked changes.

REVIEWER COMMENTS

Reviewer #1 (Remarks to the Author):

The revision has greatly improved the manuscript, and the addition of several datasets has strengthened the authors point.

I will note that I had some trouble following lines 222-231 which are fundamental to understanding the relationship between the heat budget Q_{net} and ΔQ (and thereby connection the heat budget to the mechanism described in this manuscript). Please consider rewording / reordering especially the sentence in lines 223-226.

This sentence explains the relationship between Q_{net} and ΔQ prior to the SST anomaly, do I assume correctly that after the SST anomaly Q_{net} is dominated by the warm anomaly rather than wind stress changes (and therefore ΔQ)? I assume this because there is predominantly positive correlation of ΔQ proxy precip with SST anomalies as shown in Fig 3b, but the Q_{net} becomes negative immediately as $dSST/dt$ peaks. If so please consider adding a sentence.

In the revised manuscript, we have edited the manuscript to address your comments and improved the writing and overall clarity of this section. (lines: 43-57).

It is now more clearly stated that the presence of deep atmospheric convection and diabatic heating in boreal spring, amplifies the atmosphere's sensitivity to equatorial SST anomalies and supports changes in zonal wind stress. The wind stress anomalies influences the SST anomalies by changing thermocline.

L 120 "are associated"

Corrected, thank you.

L 158-159 - does this not contradict your earlier statement that the SST anomalies are not high enough to keep the ITCZ on the equator? Or do we assume reverse causality?

The sentence is now re-written to read: “The precipitation anomalies, which are located slightly south of the climatological maximum rainfall, generally follow the northward migration of the climatological ITCZ”. (lines: 161-163).

L 238 I suggest: “After June, the atmosphere responds only weakly to SST anomalies.”
Corrected, thank you.

Reviewer #2 (Remarks to the Author):

In my original review of “Origin of the seasonality of the Atlantic Niño“

In my review I made three requests, 1) to redo the material presented in Fig1c,d to more directly relate to variations in the thermocline, 2) I requested that they actually compute the mixed layer heat budget rather than draw inferences from covariances among variables, and 3) I asked them to clarify how they handled trends in the observational data as these can impact the results. These last two issues relate to statistical confidence. With these changes, and the changes made in response to the first reviewer I am happy to recommend publication.

The revised manuscript has addressed all three of these concerns. The authors and I also had a separate exchange regarding their inclusion of a product that my group produces and they, kindly, upgraded the product that they use (which provides more consistent results).

-- Jim Carton

We very much appreciate your constructive and helpful comments. Your comments have been invaluable in revising and improving our manuscript.

REVIEWERS' COMMENTS, third round

Reviewer #1 (Remarks to the Author):

I appreciate the rewording, and recommend publication.

One minor comment to be addressed at proof stage:

l46 "across the eastern equatorial Atlantic" — should this be "across the equatorial Atlantic"?

I'm excited to see another great contribution to understanding tropical Atlantic variability published!

--Anna-Lena Deppenmeier

Response to the specific comments and suggestions from Reviewer #1:

We very much appreciate your constructive and helpful comments. The correction has been made in the manuscript and indicated in blue italic below.

Reviewer #1 (Remarks to the Author):

I appreciate the rewording, and recommend publication.

One minor comment to be addressed at proof stage:

l46 "across the eastern equatorial Atlantic" — should this be "across the equatorial Atlantic"?

I'm excited to see another great contribution to understanding tropical Atlantic variability published!

--Anna-Lena Deppenmeier

Corrected, the sentence now reads "...across the equatorial Atlantic". (line 46).